# Compound Boronizing and Its Kinetics Analysis for H13 Steel with Rare Earth CeO$_2$ and Cr$_2$O$_3$

**Shunqi Mei [1], Yanwei Zhang [1,\*], Quan Zheng [1,2,\*], Yu Fan [1], Burial Lygdenov [1,3,4,\*] and Alexey Guryev [1,3]**

[1] Hubei Provincial Key Laboratory of Digital Textile Equipment, Wuhan Textile University, Wuhan 430073, China; sqmei@wtu.edu.cn (S.M.); yfan@wtu.edu.cn (Y.F.); lygdeno59@mail.ru (A.G.)

[2] Department of Special Technology, Polzunov Altai State Technical University, 656038 Barnaul, Russia

[3] Department of Mechanical Engineering, East-Siberian State University of Technology and Management, 670013 Ulan-Ude, Russia

[4] Zhejiang Xinchang Sanxiong Bearing Co., Ltd., Shaoxing 312500, China

\* Correspondence: ywzhang@wtu.edu.cn (Y.Z.); qzheng@wtu.edu.cn (Q.Z.); lygdenov59@mail.ru (B.L.); Tel.: +86-27-59367841 (Y.Z.)

**Abstract:** The compound boronizing of H13 steel sample was carried out by adding rare earth CeO$_2$, and the effects of main parameters of compound boronizing on the properties of the boronizing layer were investigated. The boronizing kinetics were also analyzed. Under the conditions of boronizing temperatures of 850 °C, 900 °C, 950 °C, and 1000 °C, and boronizing times of 2 h, 3 h, 4 h, and 5 h, the boronizing experiment was carried out by adding CeO$_2$ with mass fraction of 2%, 4% and 6%, Cr$_2$O$_3$ with a mass fraction of 1%, 2%, 3%, and 4%. The results show that boronizing H13 steel with rare earth CeO$_2$ is conducive to the diffusion of boron atoms and can significantly improve the thickness and microhardness of boronizing layer. Boronizing kinetic analysis shows that the diffusion activation energy of boron atom decreases significantly when the rare earth content is 4%, which indicates that the content of rare earth CeO$_2$ with 4% is an ideal selection for considering improving the utilization rate of rare earth materials, and the boronizing kinetic analysis can provide a quantitative basis for determining the content of rare earth metals. Therefore, experiments and analysis show that for H13 steel, when the boronizing temperature is 950 °C, time is 4 h and rare earth content is 4%, the thickness of the boronizing layer can reach 71 μm and the microhardness at the depth of 70 μm from the surface layer can reach 1546.32 HV. Moreover, on this basis, when Cr$_2$O$_3$ with a rare earth content of 2% is added, the microstructure and wear resistance of the boronizing layer are significantly improved.

**Keywords:** compound boronization; activation energy; boronizing kinetics; rare earth CeO$_2$; H13 steel

## 1. Introduction

Boronization is a widely used thermochemical surface treatment technology to improve the friction and wear properties of metal materials. It can increase the hardness, up to 1400–2000 HV, of metal surfaces and enhance the corrosion resistance of metal surfaces, so it is more advantageous than carburizing and nitriding [1–3]. H13 steel is an air-cooling hardening hot working die steel with good strength and toughness. It is one of the most widely used in the manufacture of hot extrusion die, mandrel, hot forging die, and aluminum, copper, and its alloy die casting die. Its composition is shown in Table 1. In order to improve the surface properties of H13, quenching, tempering, ion nitriding, and other methods are generally used, but the effect of these conventional treatments often cannot meet higher and higher application requirements. Boronizing the parts of this material can improve its surface hardness and wear resistance so as to improve the service life of the die [4–6]. During the boronizing process, the structure and properties of the boronizing layer can be significantly improved by adding different elements such as rare earth, Cr, and Al for composite boronizing treatment. It has been found that the addition of rare earth can

improve the microstructure and properties of boronizing layer and increase the boronizing rate [7–9].

**Table 1.** Composition of H13 mold steel.

| Element | C | Si | Mn | Cr | Mo | V | P | S |
|---------|---|-----|-----|-----|-----|---|---|---|
| Content (%) | 0.32–0.45 | 0.80–1.20 | 0.20–0.50 | 4.75–5.50 | 1.10–1.75 | 0.80–1.20 | ≤0.030 | ≤0.030 |

To understand the boronizing kinetics is of scientific and industrial importance (for example, [9,10]). It is found that the addition of rare earth in the boronizing process can reduce the diffusion activation energy required by boron atoms during boronizing, so as to improve the effect of boronizing layer and improve boronizing efficiency. Since the radius of a rare earth atom is larger than that of an iron atom, the $\gamma$-Fe lattice is deformed and crystal defects are increased, which provides a channel for the diffusion of active boron atoms into the material and accelerates the diffusion speed of active boron atoms into the material, so as to reduce the diffusion activation energy of active boron atoms. For example, in [11,12], the addition of rare earth $Nd_2O_3$ (5%) reduced the diffusion activation energy from 184 kJ/mol to 122 kJ/mol in 45 steel and from 261 kJ/mol to 155 kJ/mol in T12 steel, which reduced the diffusion activation energy by 34% and 41% respectively, and significantly improved the boronizing efficiency. Therefore, boronizing kinetic analysis for different steel materials is a very valuable method to reveal the boronizing process.

Rare earth additives used for boronizing mainly include $CeO_2$, $La_2O_3$, $Nd_2O_3$, etc. Among them, $CeO_2$ has higher oxidation and strong adsorption performance and is easy to prepare. Therefore, $CeO_2$ is a valuable boronizing additive, which can effectively improve the microstructure of boronizing layer. Its application in boronizing with 5% rare earth($CeO_2$) of high entropy alloy AlCoCuFeMnNi can make the alloy hardness change in gradient between 260–420 $HV_{0.2}$ and stabilize at about 400 $HV_{0.2}$ [13–15]. For titanium alloy TC21, after boronizing with rare earth $CeO_2$, the wear rate of boronizing sample is 50–60 times lower than that of non boronizing sample [14,16]. In addition, adding metal elements such as aluminum (Al) and chromium ($Cr_2O_3$) during boronizing is conducive to further improve the microstructure of boronizing layer and reduce the brittleness of boronizing layer [17].

In this paper, compound boronizing with rare earth $CeO_2$ and chromium($Cr_2O_3$) for H13 steel is studied, and the effects of different proportions of rare earth $CeO_2$ in boronizing agent on the diffusion activation energy of H13 steel and the properties of boronizing layer are analyzed in order to provide a basis for further improving the properties of H13 steel.

## 2. Experiment

### 2.1. Material

The main material is H13 die steel, and its material composition is shown in Table 1.

The H13 mold steel was cut into 20 mm × 10 mm × 10 mm rectangular parallelepipeds as boronizing samples. The hardness of the H13 mold steel without boronization was 450 HV-550 HV, and the experimental boronizing paste was $B_4C$, $KBF_4$, C, bentonite, $Cr_2O_3$, $CeO_2$, $H_2O$, etc.

### 2.2. Experiment

Material treatment. Firstly, the samples shall be treated on the water mill with 180 mesh, 360 mesh, 600 mesh, and 1000 mesh sandpaper respectively, and the rust and scratches on the material surface shall be cleaned, then cleaned with alcohol and put into the drying dish for standby.

The experimental parameters are shown in Table 2. The boronizing time, temperature, and rare earth content have important effects on the boronizing results, which are the key factors investigated in this experiment.

Preparation of boronizing agent. Mix the medicine required by the boronizing agent into the mortar according to the ratio, crush the medicine, and stir it uniformly. A series of

boronizing agents were prepared by adding 2%, 4%, and 6% rare earth $CeO_2$, respectively, and 2%, 3%, 4%, and 6% $Cr_2O_3$, respectively.

**Table 2.** Experimental parameters.

| | |
|---|---|
| Paste material | $B_4C$, $KBF_4$, C, Bentonite, $CeO_2$, $Cr_2O_3$, $H_2O$, etc. |
| Paste thickness | 3–5 mm |
| Pre-treatment | Ventilated environment, 12–14 h |
| Pre-heating | In the drying oven, 200 °C, 2–3 h |
| Boriding temperature | 850 °C, 900 °C, 950 °C, 1000 °C, respectively |
| Boriding time | 2 h, 3 h, 4 h, 5 h, respectively |
| $CeO_2$ content | 0%, 2%, 4%, 6%, respectively |
| $Cr_2O_3$ content | 1%, 2%, 3%, 4%, respectively |

Compound boronizing experiment. The preparation and boronizing experiment of boronizing agent shall be carried out according to the designed test scheme.

The slurry boronizing agent is uniformly adhered to the sample surface with a thickness of 3–5 mm, and then placed in a dry and ventilated environment for 12–14 h. Then it is placed into the drying oven for preheating and drying, the temperature is set to 200 °C and the time is set to 2–3 h. Take out the sample after drying and preheating and put it into the high-temperature heating furnace for boronizing. The boronizing time is 2 h, 3 h, 4 h and 5 h, and the boronizing temperature is 850 °C, 900 °C, 950 °C and 1000 °C, respectively.

Performance test. After boronizing, the sample is taken out of the furnace for air cooling, the boronizing agent shell on the surface is removed after cooling the sample, and cut from the middle with a wire cutting machine, and one should grind, polish and corrode the cutting surface with different types of sandpaper again. The corrosion solution is 3–5% nitric acid alcohol solution [14,15]. Then, the microstructure observation, test of thickness, microhardness and wear resistance for the infiltrated layer are carried out.

The instruments are: optical microscope Olympus (DSX-HRUF), microhardness tester (HV 1000) and wear tester (MMS-2A-2). When testing the boronizing layer thickness, take 10 points to measure the thickness and calculate the average value as the measured value of boronizing layer thickness.

## 3. Results and Analysis

### 3.1. Thickness and Hardness of Boronizing Layer without Rare Earth

Table 3 shows the thickness of boronizing layer obtained in experiment for H13 steel under different temperature and time process conditions without adding rare earth (take the average value of 10 points, Hardness value of boronizing layer measured at 40 μm from the surface of boronizing layer). It can be seen from Tables 3 and 4 that when the boronizing temperature is 950 °C and the boronizing time is 4 h, the boronizing layer thickness is the largest (42 μm).

The hardness at 40 μm from the surface of the boronizing layer still reaches 1246.90 HV. Therefore, it is concluded that the ideal basic process parameters for boronizing H13 steel are: the boronizing temperature is 950 °C and the boronizing time is 4 h. This can provide a reference for further boriding experiments with rare earth.

**Table 3.** Boronizing layer thickness at different temperature and time.

| Time — Temperature | 850 °C | 900 °C | 950 °C | 1000 °C |
|---|---|---|---|---|
| 2 h | 16 μm | 23 μm | 27 μm | 28 μm |
| 3 h | 19 μm | 27 μm | 32 μm | 31 μm |
| 4 h | 20 μm | 36 μm | 42 μm | 42 μm |
| 5 h | 21 μm | 35 μm | 40 μm | 41 μm |

**Table 4.** Hardness of penetration layer at different temperature and time (40 μm from the surface of the infiltration layer).

| Time \ Temperature | 850 °C | 900 °C | 950 °C | 1000 °C |
|---|---|---|---|---|
| 2 h | 623.46 HV | 635.12 HV | 646.06 HV | 665.32 HV |
| 3 h | 612.45 HV | 626.09 HV | 723.47 HV | 684.16 HV |
| 4 h | 604.78 HV | 678.32 HV | 1246.90 HV | 1133.42 HV |
| 5 h | 632.60 HV | 667.06 HV | 1136.05 HV | 1068.30 HV |

*3.2. Thickness and Microhardness of Boronizing Layer with Rare Earth Addition*

In order to investigate the process of rare earth boronizing and analyze the boronizing kinetics, Table 5 lists the thickness of boronizing layer under different rare earth content and temperature when rare earth $CeO_2$ is added to boronizing agent and boronizing time is 4 h. Figure 1 is the variation curve of different rare earth content, boronizing temperature and boronizing layer thickness with a boronizing time of 4 h.

**Table 5.** Thickness of infiltration layer at different temperatures (Boronizing time 4 h).

| Content \ Temperature | $CeO_2$ (2%) | $CeO_2$ (4%) | $CeO_2$ (6%) |
|---|---|---|---|
| 1123 K (850 °C) | 22 μm | 31 μm | 33 μm |
| 1173 K (900 °C) | 42 μm | 57 μm | 58 μm |
| 1223 K (950 °C) | 54 μm | 71 μm | 74 μm |
| 1273 K (1000 °C) | 59 μm | 75 μm | 79 μm |

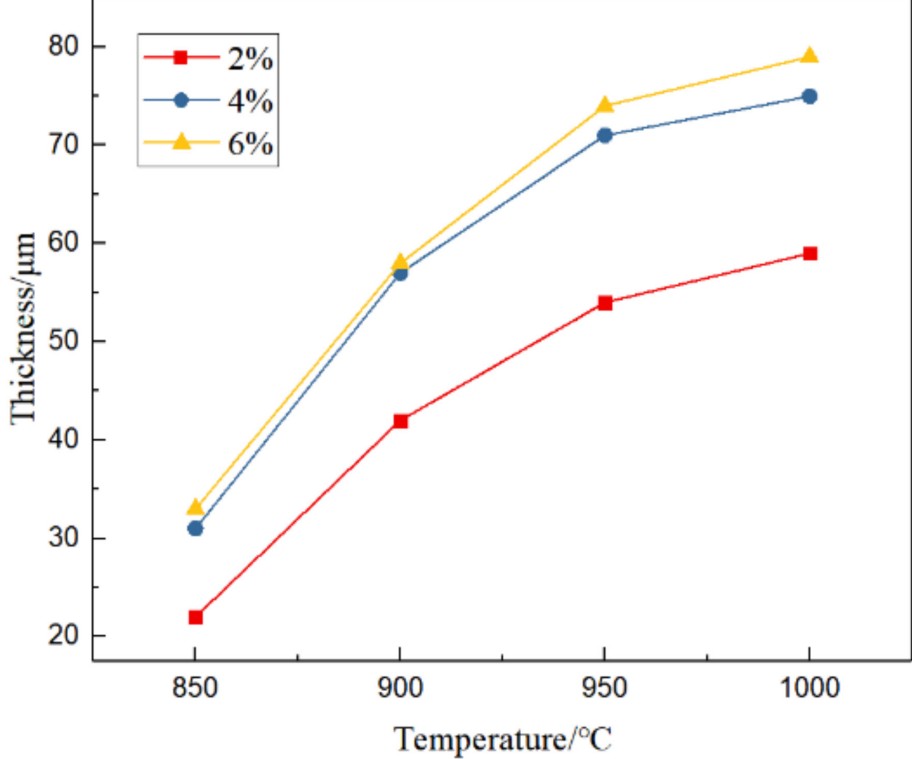

**Figure 1.** The increase of thickness of boronizing layer with solution and temperature (4 h).

From the comparison between Tables 3 and 5, it is easy to find that the boronizing layer thickness increases greatly with the addition of rare earth $CeO_2$. For example, when the boronizing temperature is 950 °C and the boronizing time is 4 h, the thickness of the boronizing layer increases from 42 μm when rare earth is not added to 71 μm when rare earth is 4%.

It can be seen from Table 5 and Figure 1 that the boronizing layer thickness increases gradually with the increase of temperature, and at the same temperature, and at the same temperature, with the increase of the content of rare earth $CeO_2$, the thickness of the boronizing layer also shows an increasing trend. This shows that the increase of rare earth content can promote the infiltration of boron atoms into the bulk material and increase the thickness of boronizing layer [18].

For example, when the boronizing temperature is 950 °C (1223 K), when the rare earth content increases from 2% to 4%, the boronizing layer thickness increases significantly from 54 μm to 71 μm (up to 31.5%).

However, when the rare earth content increases from 4% to 6%, the increase of infiltration layer thickness is not obvious (From 71 μm to 74 μm, the increase is only 4.2%). This phenomenon will be explained by the boronizing kinetic analysis below.

Table 6 shows the microhardness of the boronizing layer with different rare earth $CeO_2$ content (2%, 4% and 6%) at the boronizing temperature of 950 °C and boronizing time of 4 h. Figure 2 shows the microhardness curve of the boronizing layer. It can be seen from Table 6 and Figure 2 that the microhardness of each group when the content of rare earth $CeO_2$ is 4% is greater than that when the content is 2% and 6%.

**Table 6.** Boronizing layer hardness (boronizing temp. 950 °C, time 4 h).

| Surface Depth | ($CeO_2$) 2% Hardness/HV | ($CeO_2$) 4% Hardness/HV | ($CeO_2$) 6% Hardness/HV |
|---|---|---|---|
| 20 μm | 1559.93 | 1571.97 | 1411.97 |
| 40 μm | 1412.10 | 1529.93 | 1509.04 |
| 60 μm | 714.29 | 1670.02 | 1173.15 |
| 70 μm | 662.47 | 1546.32 | 1125.45 |
| 100 μm | 686.60 | 652.42 | 624.31 |

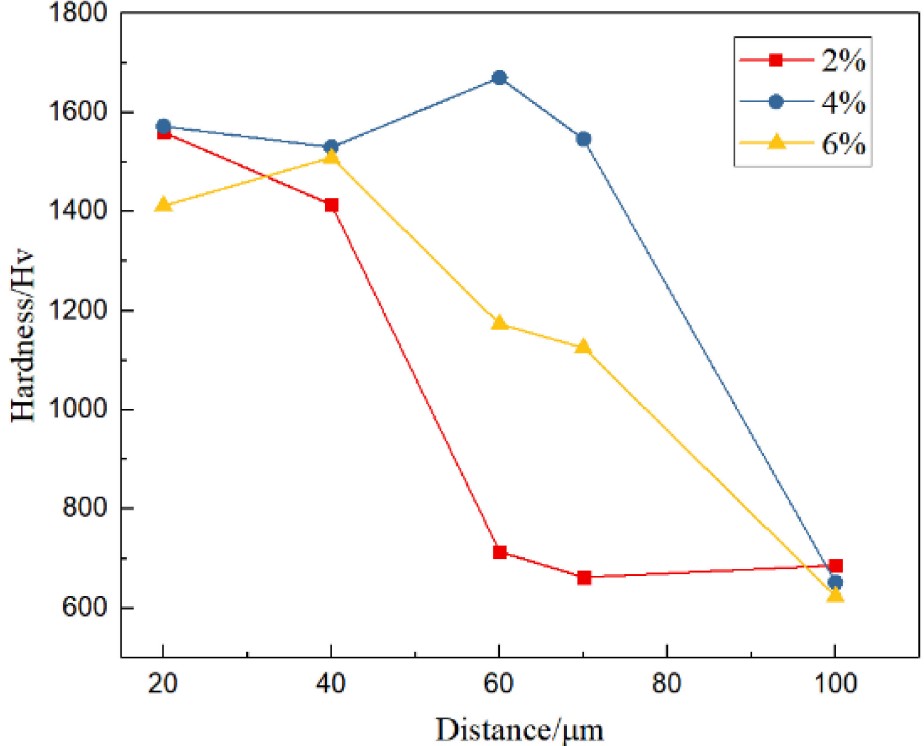

**Figure 2.** Hardness curves of different rare earth $CeO_2$ contents.

When the content of rare earth is 4%, the hardness at 70 μm from the surface layer reaches 1546.32 HV, which is significantly higher than that at the same depth when the content is 2% and 6% (662.47 HV and 1125.45 HV, respectively). At the same time, when the rare earth content is 4%, there is little difference in hardness from the surface to 70 μm

away from the surface of the boronizing layer. This shows that when the content of rare earth $CeO_2$ is 4%, the hardness of boronizing layer is high and relatively stable [18].

The analysis of the above results preliminarily shows that when adding rare earth $CeO_2$ for boronizing, the ideal boronizing layer thickness and microhardness can be obtained when the boronizing temperature is 950 °C, the boronizing time is 4 h, and the rare earth content is 4%.

### 3.3. Boronization Kinetics Analysis

According to the Arrhenius equation, the relationship between the penetration time, the thickness of the penetration layer, and the growth coefficient can be written as the following Equation (1) [16].

$$X^2 = Gt \tag{1}$$

where $X$ is the thickness of the boronizing layer (μm), $G$ is the growth coefficient (m²·s), and $t$ is the boronization time (s).

From the Arrhenius equation can get Equation (2) [16], where $T$ is the absolute temperature (K), $Q$ is the diffusion activation energy (J mol), $R$ is the gas constant (J·(mol·k)$^{-1}$), and $G_0$ is the diffusion constant (m²·s),

$$\ln G = \ln G_0 - \frac{Q}{R}\frac{1}{T} \tag{2}$$

From Equations (1) and (2), the growth coefficient and diffusion activation energy at different rare earth can be obtained as shown in Table 7, The growth coefficients under different rare earth $CeO_2$ contents can be obtained from Equations (1) and (2), as shown in Table 7. The relation curve of diffusion activation energy $LnG$ and $1/T$ is shown in Figure 3, where K represents the absolute temperature (K1123 = 850 °C, K1173 = 900 °C, K1223 = 950 °C, K1273 = 1000 °C).

**Table 7.** Growth factors at different rare earth ($CeO_2$) contents.

| Condition | Growth Factor (m²·s) | | | |
|---|---|---|---|---|
| 2% $CeO_2$ | $K_{1123} = 3.36 \times 10^{-14}$ | $K_{1173} = 1.23 \times 10^{-13}$ | $K_{1223} = 2.03 \times 10^{-13}$ | $K_{1273} = 2.42 \times 10^{-13}$ |
| 4% $CeO_2$ | $K_{1123} = 6.67 \times 10^{-14}$ | $K_{1173} = 2.26 \times 10^{-13}$ | $K_{1223} = 3.5 \times 10^{-13}$ | $K_{1273} = 3.91 \times 10^{-13}$ |
| 6% $CeO_2$ | $K_{1123} = 7.56 \times 10^{-14}$ | $K_{1173} = 2.34 \times 10^{-13}$ | $K_{1223} = 3.8 \times 10^{-13}$ | $K_{1273} = 4.33 \times 10^{-13}$ |

It can be seen from Figure 3 that the linear slope at 2% of rare earth content is greater than 4%, and the slope at 4% is almost parallel to that at 6%, indicating that the diffusion activation energy of boron atom changes when the rare earth content increases from 2% to 4%, while there is basically no change from 4% to 6%. Table 8 shows the activation energy of different contents of rare earth $CeO_2$, 160.70 kJ/mol at 2%, 143.94 kJ/mol at 4% and 143.16 kJ/mol at 6%. It can be found from Table 7 that when the content of rare earth $CeO_2$ increases from 2% to 4%, the diffusion activation energy decreases by 16.76 kJ/mol (relative decrease of 10.4%), and when the content of rare earth $CeO_2$ increases from 4% to 6%, the diffusion activation energy only decreases by 0.78 kJ/mol (relative decrease of 0.54%), which is not obvious [18].

**Table 8.** Reduction of activation energy with different contents of rare earth $CeO_2$.

| | $CeO_2$ 2% | $CeO_2$ 4% | $CeO_2$ 6% |
|---|---|---|---|
| Diffusion activation energy (kJ/mol) | 160.70 | 143.94 | 143.16 |
| Relatively reduced diffusion activation energy (kJ/mol) | Contrast value | 16.76 | 0.78 |

It shows that when rare earth $CeO_2$ increases from 4% to 6%, the effect of further promoting boron atom diffusion is very small, so the increase of infiltration layer thickness is also very small. This also shows that the utilization rate of rare earth decreases significantly when the content of rare earth is further increased.

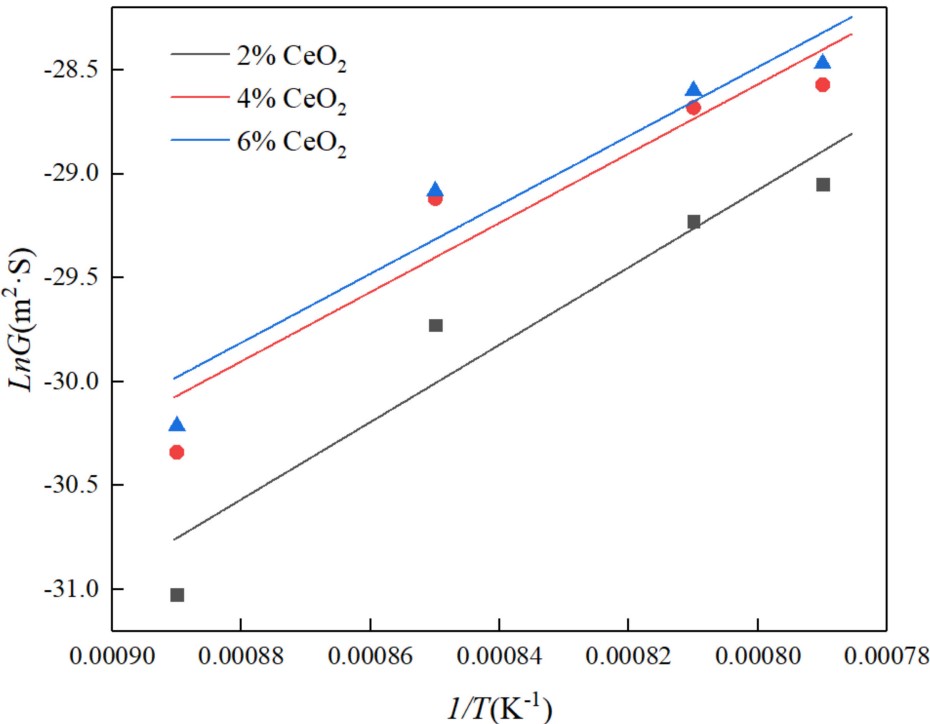

**Figure 3.** Relationship between Ln*G* and $1/T$ under different rare earth contents (straight line).

Therefore, this theoretically shows that the content of rare earth $CeO_2$ of 4% is an ideal proportion.

### 3.4. Microstructure of Rare Earth Boron-Chromium Co-Infiltration

In our previous studies, it was found that when the $CeO_2$ rare earth content was 4%, the boronizing layer had better combination and performance [18]. Now, we studied the effect of different $Cr_2O_3$ content on rare earth boron-chromium co-infiltration. Figure 4 shows the microstructure of boronizing layer with $Cr_2O_3$ content of 1%, 2%, 3%and 4% respectively at boronizing temperature of 950 °C, time of 4 h, and rare earth content of 4%. As can be seen from Figure 4, with the increase of $Cr_2O_3$ content, the microstructure of boronizing layer is gradually changing. When the content is 1%, the microstructure of boronizing layer is relatively sparse, and a large number of holes appear in boronizing layer. At 2%, the boronizing microstructure is compact and there are few holes; at 3%, 4%, the microstructure of boronizing layer shows a few holes. This shows that too much chromium oxide ($Cr_2O_3$) is added to the boronizing agent, which may cause chromium atoms to accumulate on the surface of the boronizing layer, thereby inhibiting the growth of the boronizing layer and reducing the thickness of the boronizing layer. The experiment also shows that the boronizing layer has better microstructure when the content of $Cr_2O_3$ is 2%.

Figure 5 shows the structure of the rare-earth boron-chromium co-diffusion layer at different temperatures when boronizing time is 4 h, rare earth $CeO_2$ content is 4% and $Cr_2O_3$ content is 2%. The figure shows that as the boronizing temperature increases from 850 °C to 950 °C, the boronizing layer structure becomes continuous and compact, However, when the boronizing temperature increased from 950 °C to 1000 °C, holes began to appear in the structure of the boronizing layer. When the boronizing temperature exceeded 980 °C, holes appeared in the outermost layer of the boronizing layer. This is due to the fact that the temperature rises too high, accelerates the diffusion of boron atoms, and it is difficult for boron atoms to form a continuous boronizing layer with the surface Fe atoms. The increase in temperature also causes more gas to escape from the infiltrated layer, resulting in holes and sparseness on the surface of the boronizing layer. The results show that the

boronizing temperature of 950 °C is a reasonable boronizing temperature for the rare-earth boron-chromium co-diffusion.

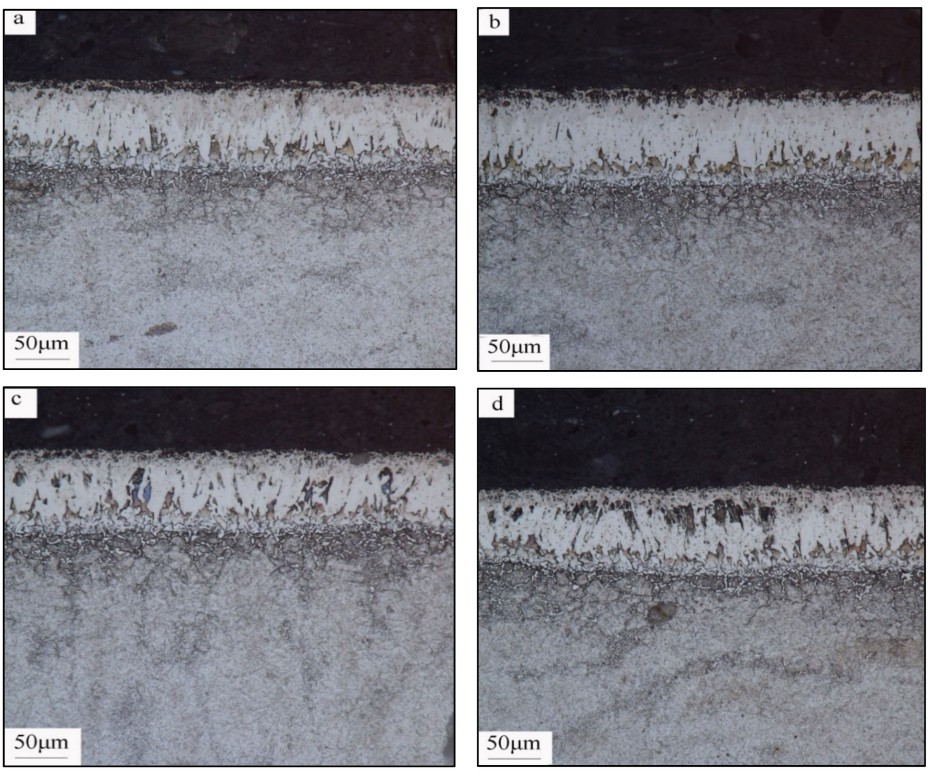

**Figure 4.** Microstructure of boronizing layer with $Cr_2O_3$ content of 1%, 2%, 3%, and 4%, respectively. (**a**) 1% $Cr_2O_3$; (**b**) 2% $Cr_2O_3$; (**c**) 3% $Cr_2O_3$; and (**d**) 4% $Cr_2O_3$.

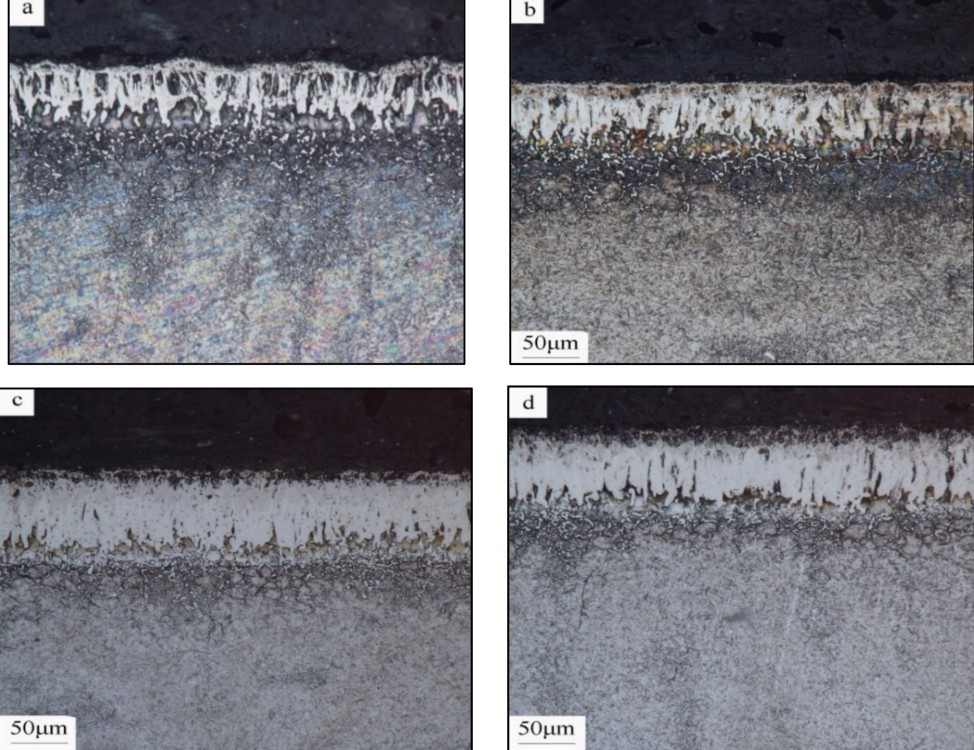

**Figure 5.** Microstructure of boronizing layer at different boronizing temperatures (boronizing time 4 h, rare earth content 4%, $Cr_2O_3$ content 2%). (**a**) 850 °C; (**b**) 900 °C; (**c**) 950 °C; and (**d**) 1000 °C.

*3.5. Wear Resistance Test of Boronizing Layer*

Table 9 shows the mass loss of the boronizing samples in friction and wear test for the three cases (single boronizing, rare earth boronizing, rare earth boron-chromium co-boriding). Test force: 100 N, rotating speed: 200 r/min, test time: 60 min, record the sample mass loss data every 10 min. It can also be seen from Table 8 that the mass loss of the single boronizing sample is the largest, and that of the rare earth boron-chromium co-boriding is the smallest.

**Table 9.** Mass loss of boronizing samples for the three cases.

| Wear Time/min | Mass Loss of Single Boronizing Sample/mg | Mass Loss of Boronizing Sample with Rare Earth Content of 4%/mg | Mass Loss of Boronizing Sample with Chromium Oxide Content of 2% and Rare Earth of 4%/mg |
|---|---|---|---|
| 10 | 4.8 | 1.2 | 0.8 |
| 20 | 6.6 | 2.5 | 1.7 |
| 30 | 9.2 | 3.8 | 2.7 |
| 40 | 12.1 | 5.1 | 3.6 |
| 50 | 16.7 | 6.9 | 4.8 |
| 60 | 22.3 | 9.2 | 6.1 |

Among them, in the 60 min wear time, the mass loss of single boronizing is 22.3 mg, that of rare earth boronizing is 9.2 mg, and that of rare earth boron-chromium co-infiltration is 6.1 mg. Among them, the mass loss of single boronization is 3.7 times that of boron-chromium rare earth co-infiltration, and 2.4 times that of rare earth boronization, which shows that the wear resistance of rare earth boron-chromium co-infiltration is relatively better. This is due to the fact that the addition of rare earth increases the activity of boron atoms, which provides a channel for the active boron atoms to diffuse into the material, thereby improving the boronizing efficiency. And at the same time, $Cr_2O_3$ strengthens the metal bonds in the local area of the B-B atomic bond, which improves the quality of the boronizing layer. is improved, thereby improving the wear resistance of the boronizing layer.

**4. Conclusions**

This paper shows that the compound boronizing of H13 steel with rare earth $CeO_2$ is conducive to the diffusion of boron atoms, and can significantly improve the thickness, microhardness, microstructure and wear resistance of boronizing layer. For example, when the boronizing temperature is 950 °C and the boronizing time is 4 h, the thickness of the boronizing layer increases from 42 μm to 71 μm. The microhardness at the depth of 70 μm from the surface layer can reach 1546.32 HV. This shows that not only the thickness of the boronizing layer is increased, but also the hardness consistency in the boronizing layer is good, which can effectively improve the surface properties and service life of the boronizing layer.

The kinetic analysis of rare earth boronizing of H13 steel shows that when the content of rare earth $CeO_2$ increases from 2% to 4%, the diffusion activation energy decreases by 16.76 kJ/mol (relative decrease of 10.4%), and when the content increases from 4% to 6%, the diffusion activation energy changes very little, only 0.54%. Boronizing kinetic analysis can provide quantitative basis for determining the content of rare earth. The study shows that adding 2% $Cr_2O_3$ can obtain better microstructure and wear resistance, since rare earth and $Cr_2O_3$ can provide more diffusion channels and strengthen the B-B atomic bond. The research methods and results of this paper can provide reference for rare earth boron-chromium co-infiltration process of carbon steel. Therefore, in order to improve the utilization rate of rare earth and $Cr_2O_3$ materials, the content of rare earth $CeO_2$ of 4% and $Cr_2O_3$ of 2% is an ideal proportion. The theoretical analysis and experimental results show that the boronizing kinetic analysis can provide a quantitative basis for determining the

content of rare earth. The research methods and results of this paper can provide reference for rare earth boron-chromium co-infiltration process of carbon steel.

**Author Contributions:** Conceptualization, S.M., Y.Z., B.L. and A.G.; methodology, S.M., Y.Z. and Q.Z.; software, Y.Z., Q.Z. and Y.F.; validation, S.M., Y.Z.; formal analysis, Y.Z., Q.Z. and Y.F.; investigation, Q.Z., Y.F.; resources, S.M.; data curation, Y.Z., Y.F.; writing—original draft preparation, Y.Z., S.M.; writing—review and editing, S.M., Y.Z.; visualization, S.M., Y.Z.; supervision, S.M.; project administration, S.M.; funding acquisition, S.M. All authors have read and agreed to the published version of the manuscript.

**Funding:** This work was supported by The Science and Technology Program of Hubei Province (No.2019AEE011, 2018AAA036) and The National Science Foundation of China of China (No.51175385).

**Institutional Review Board Statement:** The study did not involve humans or animals.

**Informed Consent Statement:** The study did not involve humans.

**Data Availability Statement:** Data is contained within the article.

**Conflicts of Interest:** The authors declare no conflict of interest. The funders had no role in the design of the study; in the collection, analyses, or interpretation of data; in the writing of the manuscript, or in the decision to publish the results.

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
