# Peer review of "Compound Boronizing and Its Kinetics Analysis for H13 Steel with Rare Earth CeO2 and Cr2O3"

_applsci, doi:10.3390/app12073636_

Round 1
Reviewer 1 Report
- The subject of the article is of current interest following the journal's profile.
- The research undertaken by the article's authors seems to be interesting. It could certainly involve considerable effort, but the way it is expressed in English seems to be negligent, including in terms of the requirements for drafting the text of a scientific article.
- Thus, the concept of "compound boronizing" does not seem to be found in the literature.
- In the Abstract, the authors show "The boronizing compound of H13 steel surface", as if H13 steel had a surface. It could be the surface of a test sample or a part of H13 steel, but boronizing does not mean the deposition of boron on the surface of the test sample, but the diffusion of boron in the surface layer of the test sample or part.
- Also, in the Abstract, the authors show that "Therefore, experiments and analysis show that for H13 steel, boronizing temperature of 950 ℃, time of 4 hours and rare earth content of 4% are ideal boronizing parameters". The statement is far too categorical, not being sufficiently justified even in the actual text of the article. Some values ​​of the boronizing parameters may lead to some somewhat convenient results, but without reaching this conclusion as a result of an appropriate scientific approach.
- There are many situations in the article in which no space was left between the numbers and the units of measurement ("71m", instead of "71 m", "200HV", instead of "200 HV", etc.). Blank spaces are also missing from other words in the text of the article (for example, "and chromium(Cr2O3)during", "CeO2 and chromium(Cr2O3)for", etc.).
The '/' sign does not require spaces before and after it (" 184 kJ/mol" must be written instead of" 184 kJ / mol"). It is necessary to write "boronizing efficiency" instead of "boronizing efficienc" etc.
- In subchapter 2.1, there is the wording "and the experimental boronizing paste was C, B4C, rare earth cerium oxide CeO2". The boronizing paste included the substances mentioned and was not what the authors mentioned.
- Sometimes, the mode of expression is rather specific to imperative recommendations to the reader, not being assimilated to the mode of scientific expression (e.g., "mix the ..., crush the ..., and stir ...", "Take out the sample ...", etc.).
- In Subchapter 2.2, a less common method in scientific articles is also used, introducing a concept followed by a colon and then some clarifications (e.g., "Performance test: after boronizing, take the sample ...". After the colon, sometimes the authors started the first word in lower case, sometimes in upper case.
- The wording "It can be seen from table 5 and figure 1 that the boronizing layer thickness increases gradually with the increase of temperature, and at the same temperature, the boronizing layer thickness when the content of rare earth CeO2 is 6% > the boronizing layer thickness when the content is 4% > the boronizing layer thickness when the content is 2%." is long, confusing and does not respect the conventions of scientific expression. It is necessary to use short and clear sentences.
There are many confusing expressions in the text of the paper (for example, "Figure 5 (a), (b), (c) and (d) are the microstructure of boronized layer", on page 8, etc.).
- In the case of the wording "the thickness of the boronizing layer increases from 42m when rare earth is not added to 71m 140 when rare earth is 4%.", the units of measurement for the thickness of the boronized layer are wrong.
- The symbol "k" used to express the absolute temperature "(1223k)" is misspelled (using the lower case letter).
- Uppercase and lowercase letters are used incorrectly in the case of the legend of Figure 1 ("Figure 1. the increase of Thickness of boronized ...").
- The explanatory text after equation (1) should start at the left zone of the line and with a lower case letter.
- In the text of the paper, the symbols of different sizes should be written in italics.
- Equations (1) and (2) have been taken from the reference [16]. There is a reference to the bibliographic reference [16] only for Equation (1). For Equation (2), no reference to the bibliographic reference [16], has been included, although this Equation is also found in the bibliographic reference [16].
- The symbol used to express the unit of measurement of rotating speed on page 9 ("rotating speed: 200R/min") is not the usual one.
- In the case of the expression "The weight loss of boronizing sample with 2% rare earth CeO2 is 16.3mg", it is "mass lost", for which the unit of measurement "mg" is valid.
- In the list of bibliographic references, it is preferable to include all the authors of the cited works.
- The authors do not appear to have complied with the valid requirements for writing the list of bibliographic references (abbreviated journal titles were not used, the volume number is not written in italics, the punctuation mark appears "colon" before the page numbers, etc.).
- The authors appear to have used the same equations and some photographs from Zhang Yanwei, Zheng Quan, Fan Yu, Mei Shunqi, Lygdenov Burial, Guryev Alexey, Effects of CeO2 content, boronizing temperature and time on microstructure and properties of H13 steel boronized layer, Materials for Mechanical Engineering, 2021, Vol. 45, Issue 7, 22-26, DOI: 10.11973/jxgccl202107005, available at http://qikan.cmes.org/EN/abstract/abstract53640.shtml#. This paper is not mentioned in the list of bibliographic references, and as such, there are no bibliographic references when there are issues that are found in both papers. Authors should replace the figures used in the paper proposed for publication or refer to the paper in which they published the same results in terms of figures, equations, etc.
- Another recommendation is to have the article corrected by someone more fluent in English and possibly more familiar with the current conventions of writing a scientific article. It is also necessary to follow the recommendations in the Instructions for Authors (https://www.mdpi.com/journal/applsci/instructions) (for example, see how to write Figure and Table words, which should start with a capital letter).
In its current form, the article contains too many English expression errors and scientific language errors, respectively, even if the research undertaken is interesting and useful.
Author Response
Thank you for your letter and for the reviewers’ comments concerning our manuscript entitled “Compound boronizing and its kinetics analysis for H13 steel with rare earth CeO2” Those comments are all valuable and very helpful for revising and improving our paper, as well as the important guiding significance to our researches. We have studied comments carefully and have made correction which we hope meet with approval. Revised portion are marked in red in the paper. The main corrections in the paper and the responds to the reviewer’s comments are as flowing:
- The subject of the article is of current interest following the journal's profile.
Response:Thank you for your approval.
2.The research undertaken by the article's authors seems to be interesting. It could certainly involve considerable effort, but the way it is expressed in English seems to be negligent, including in terms of the requirements for drafting the text of a scientific article.
Response: We are very sorry for our incorrect writing,We have invited those who are more fluent in English and may be more familiar with the current practice of writing scientific articles to correct the English expression of the articles.
3.Thus, the concept of "compound boronizing" does not seem to be found in the literature.
Response: We have made correction according to the Reviewer’s comments. In the article we have added the concept of composite boronizing.
4.In the Abstract, the authors show "The boronizing compound of H13 steel surface", as if H13 steel had a surface. It could be the surface of a test sample or a part of H13 steel, but boronizing does not mean the deposition of boron on the surface of the test sample, but the diffusion of boron in the surface layer of the test sample or part.
Response: We have made correction according to the Reviewer’s comments as: “The compound boronizing of H13 steel sample was carried out…… ”
5.Also, in the Abstract, the authors show that "Therefore, experiments and analysis show that for H13 steel, boronizing temperature of 950 ℃, time of 4 hours and rare earth content of 4% are ideal boronizing parameters". The statement is far too categorical, not being sufficiently justified even in the actual text of the article. Some values of the boronizing parameters may lead to some somewhat convenient results, but without reaching this conclusion as a result of an appropriate scientific approach.
Response: We have re-written this part according to the Reviewer’s suggestion , Amend the sentence in the abstract to read : When the boronizing temperature of 950 ℃, time of 4 hours and rare earth content of 4 %, the thickness of boronizing layer can reach 71μm, the microhardness at the depth of 70 μm from the surface layer can reach 1546.32 HV.
- There are many situations in the article in which no space was left between the numbers and the units of measurement ("71m", instead of "71 m", "200HV", instead of "200 HV", etc.). Blank spaces are also missing from other words in the text of the article (for example, "and chromium(Cr2O3)during", "CeO2 and chromium(Cr2O3)for", etc.).
The '/' sign does not require spaces before and after it (" 184 kJ/mol" must be written instead of" 184 kJ / mol"). It is necessary to write "boronizing efficiency" instead of "boronizing efficienc" etc.
Response: In the article we fixed all unit formatting errors, we have re-written this part according to the Reviewer’s suggestion. added space between numbers and units of measure.
7.In subchapter 2.1, there is the wording "and the experimental boronizing paste was C, B4C, rare earth cerium oxide CeO2". The boronizing paste included the substances mentioned and was not what the authors mentioned.
Response: We now amend this to read: the experimental boronizing paste was B4C, KBF4, C, Bentonite,Cr2O3, CeO2 , H2O etc. Among them, B4C, KBF4, C, and Bentonite basic boronizing components.
8.Sometimes, the mode of expression is rather specific to imperative recommendations to the reader, not being assimilated to the mode of scientific expression (e.g., "mix the ..., crush the ..., and stir ...", "Take out the sample ...", etc.).
Response: We are very sorry for our incorrect writing , We invited corrections to the article from those who are more fluent in English and who may be more familiar with current scientific article writing practices.
9.In Subchapter 2.2, a less common method in scientific articles is also used, introducing a concept followed by a colon and then some clarifications (e.g., "Performance test: after boronizing, take the sample ...". After the colon, sometimes the authors started the first word in lower case, sometimes in upper case.
Response: We are very sorry for our incorrect writing , We have changed all the colons in the article to full stops and have standardised the initials to capital letters.
- We have re-written this part according to the Reviewer’s suggestion. The wording "It can be seen from table 5 and figure 1 that the boronizing layer thickness increases gradually with the increase of temperature, and at the same temperature, the boronizing layer thickness when the content of rare earth CeO2 is 6% > the boronizing layer thickness when the content is 4% > the boronizing layer thickness when the content is 2%." is long, confusing and does not respect the conventions of scientific expression. It is necessary to use short and clear sentences.
There are many confusing expressions in the text of the paper (for example, "Figure 5 (a), (b), (c) and (d) are the microstructure of boronized layer", on page 8, etc.).
Response: The question in the article has been modified, in the article we have used a more scientific, shorter and clearer expression to describe.
11.In the case of the wording "the thickness of the boronizing layer increases from 42m when rare earth is not added to 71m 140 when rare earth is 4%.", the units of measurement for the thickness of the boronized layer are wrong.
Response: We are very sorry for our incorrect writing .The unit is changed to“μm”.
12.The symbol "k" used to express the absolute temperature "(1223kK)" is misspelled (using the lower case letter).
Response: We are very sorry for our incorrect writing . All absolute temperature symbols are modified to "K".
13.Uppercase and lowercase letters are used incorrectly in the case of the legend of Figure 1 ("Figure 1. the increase of Thickness of boronized ...").
Response: We are very sorry for our incorrect writing , we have modified this to read: :Figure 1. The increase of thickness of boronized …
14.The explanatory text after equation (1) should start at the left zone of the line and with a lower case letter.
Response: We have re-written this part according to the Reviewer’s suggestion. The explanatory text after equation (1) already start at the left zone of the line and with a lower case letter.
15.In the text of the paper, the symbols of different sizes should be written in italics.
Response: We have re-written this part according to the Reviewer’s suggestion, edited to italic
16.Equations (1) and (2) have been taken from the reference [16]. There is a reference to the bibliographic reference [16] only for Equation (1). For Equation (2), no reference to the bibliographic reference [16], has been included, although this Equation is also found in the bibliographic reference [16].
Response: Thanks for your reminder. We have given the reference for Equation (2), reference to the bibliographic reference [16].
17.The symbol used to express the unit of measurement of rotating speed on page 9 ("rotating speed: 200R/min") is not the usual one.
Response: We have re-written this part according to the Reviewer’s suggestion, Modified to: 200 r / min.
18.In the case of the expression "The weight loss of boronizing sample with 2% rare earth CeO2 is 16.3mg", it is "mass lost", for which the unit of measurement "mg" is valid.
Response: Thanks for your reminder. We have re-written this part according to the Reviewer’s suggestion, all “weight loss”is modified to “mass loss”.
19.In the list of bibliographic references, it is preferable to include all the authors of the cited works.
Response: Thanks for your reminder. We have re-written this part according to the Reviewer’s suggestion. References include all the authors of the cited works.
20.The authors do not appear to have complied with the valid requirements for writing the list of bibliographic references (abbreviated journal titles were not used, the volume number is not written in italics, the punctuation mark appears "colon" before the page numbers, etc.).
Response: Thanks for your reminder. We have re-written this part according to the Reviewer’s suggestion. Amendment has complied with valid requirements for writing a bibliography list.
21.The authors appear to have used the same equations and some photographs from Zhang Yanwei, Zheng Quan, Fan Yu, Mei Shunqi, Lygdenov Burial, Guryev Alexey, Effects of CeO2 content, boronizing temperature and time on microstructure and properties of H13 steel boronized layer, Materials for Mechanical Engineering, 2021, Vol. 45, Issue 7, 22-26, DOI: 10.11973/jxgccl202107005, available at http://qikan.cmes.org/EN/abstract/abstract53640.shtml#. This paper is not mentioned in the list of bibliographic references, and as such, there are no bibliographic references when there are issues that are found in both papers. Authors should replace the figures used in the paper proposed for publication or refer to the paper in which they published the same results in terms of figures, equations, etc.
Response: Thanks for your comments. We have re-written this part according to the Reviewer’s suggestion . The photos and contents of Figure 4, figure 5 and table 9 have been replaced with the photos and contents of the research of this paper. At the same time, individual contents cite the articles we have published "Zhang Yanwei, Zheng Quan, Fan Yu, Mei Shunqi, Lygdenov Burial, Guryev Alexey, Effects of CeO2 content, boronizing temperature and time on microstructure and properties of H13 steel boronized layer, Materials for Mechanical Engineering, 2021, Vol. 45, Issue 7, 22-26, DOI: 10.11973/jxgccl202107005", We have mentioned this article in the list of bibliographic references, and have given bibliographic references in appropriate places.
22.Another recommendation is to have the article corrected by someone more fluent in English and possibly more familiar with the current conventions of writing a scientific article. It is also necessary to follow the recommendations in the Instructions for Authors (https://www.mdpi.com/journal/applsci/instructions) (for example, see how to write Figure and Table words, which should start with a capital letter).
Response: We are very sorry for our incorrect writing,we have now made corrections to those who are more fluent in English and who may be more familiar with current scientific article writing conventions.
Special thanks to you for your good comments.

Reviewer 2 Report
The authors have investigated the boronizing of H13 steel with rare earth as additives. The effects of processing temperature, boronizing time and rare earth content on the properties of the boronized layer have been compared to retrieve the optimal boronizing parameters. The experiment is well conducted. However, some information and details are not present very well in the discussion. The manuscript can be reconsidered for publication after major revision.
- Lines 117-119 are the same as lines 120-122, consider removing or rephrasing.
- Line 160, table 5 and figure 3 are not correctly linked to the content discussed here.
- Lines 252-254 are the same as lines 249-251, consider removing or rephrasing.
- Line 246, table 8 is not correctly linked to the content discussed here.
- The authors have measured the properties of different boronized layers with varying boronizing parameters. However, those results were not compared to literature data before drawing the conclusion that 4% CeO2 is the ideal selection. It would be better to include results from similar studies in order to evidence the scientific novelty of the present paper.
Author Response
Thank you for your letter and for the reviewers’ comments concerning our manuscript entitled “Compound boronizing and its kinetics analysis for H13 steel with rare earth CeO2” Those comments are all valuable and very helpful for revising and improving our paper, as well as the important guiding significance to our researches. We have studied comments carefully and have made correction which we hope meet with approval. Revised portion are marked in red in the paper. The main corrections in the paper and the responds to the reviewer’s comments are as flowing:
- Lines 117-119 are the same as lines 120-122, consider removing or rephrasing.
Response: The question in the article has been modified, in the article we have used a more scientific, shorter and clearer expression to describe
- Line 160, table 5 and figure 3 are not correctly linked to the content discussed here.
Response:We are very sorry for our incorrect writing, change into :table 5 and figure 1.
- Lines 252-254 are the same as lines 249-251, consider removing or rephrasing.
Response: The question in the article has been modified, in the article we have used a more scientific, shorter and clearer expression to describe
- Line 246, table 8 is not correctly linked to the content discussed here.
Response:We are very sorry for our incorrect writing, change into :table 5 and figure 9.
5.The authors have measured the properties of different boronized layers with varying boronizing parameters. However, those results were not compared to literature data before drawing the conclusion that 4% CeO2 is the ideal selection. It would be better to include results from similar studies in order to evidence the scientific novelty of the present paper.
A: We have made correction according to the Reviewer’s comments. We have added literature data for comparison.
Special thanks to you for your good comments.

Round 2
Reviewer 2 Report
Dear Editor,
The authors have addressed all the issues raised in the previous review. The manuscript can now be considered for publication in Applied Sciences.
Please do not hesitate to contact me if you have any questions.
Sincerely,
Xialu Wei